# Estimations of Mutation Rates Depend on Population Allele Frequency Distribution: The Case of Autosomal Microsatellites

**DOI:** 10.3390/genes13071248

**Published:** 2022-07-14

**Authors:** Sofia Antão-Sousa, Eduardo Conde-Sousa, Leonor Gusmão, António Amorim, Nádia Pinto

**Affiliations:** 1Instituto de Investigação e Inovação em Saúde (i3S), 4200-135 Porto, Portugal; econdesousa@gmail.com (E.C.-S.); aamorim@ipatimup.pt (A.A.); npinto@ipatimup.pt (N.P.); 2Institute of Molecular Pathology and Immunology, University of Porto (IPATIMUP), 4200-465 Porto, Portugal; 3Faculty of Sciences, University of Porto (FCUP), 4169-007 Porto, Portugal; 4DNA Diagnostic Laboratory (LDD), State University of Rio de Janeiro (UERJ), Rio de Janeiro 20550-013, Brazil; leonorbgusmao@gmail.com; 5Instituto de Engenharia Biomédica (INEB), 4200-135 Porto, Portugal; 6Center of Mathematics, University of Porto (CMUP), 4169-007 Porto, Portugal

**Keywords:** microsatellites, STRs, autosomes, mutation rate estimates biases, hidden mutations, dating, evolution

## Abstract

Microsatellites (or short-tandem repeats (STRs)) are widely used in anthropology and evolutionary studies. Their extensive polymorphism and rapid evolution make them the ideal genetic marker for dating events, such as the age of a gene or a population. This usage requires the estimation of mutation rates, which are usually estimated by counting the observed Mendelian incompatibilities in one-generation familial configurations (typically parent(s)–child duos or trios). Underestimations are inevitable when using this approach, due to the occurrence of mutational events that do not lead to incompatibilities with the parental genotypes (‘hidden’ or ‘covert’ mutations). It is known that the likelihood that one mutation event leads to a Mendelian incompatibility depends on the mode of genetic transmission considered, the type of familial configuration (duos or trios) considered, and the genotype(s) of the progenitor(s). In this work, we show how the magnitude of the underestimation of autosomal microsatellite mutation rates varies with the populations’ allele frequency distribution spectrum. The Mendelian incompatibilities approach (MIA) was applied to simulated parent(s)/offspring duos and trios in different populational scenarios. The results showed that the magnitude and type of biases depend on the population allele frequency distribution, whatever the type of familial data considered, and are greater when duos, instead of trios, are used to obtain the estimates. The implications for molecular anthropology are discussed and a simple framework is presented to correct the naïf estimates, along with an informatics tool for the correction of incompatibility rates obtained through the MIA.

## 1. Introduction

Microsatellites (or short tandem repeats (STRs)) have been widely used in a wide range of scientific fields, such as population and forensic genetics, anthropology, and evolution; see, for example, [1,2,3]. For most of these usages, a critical parameter is required: the (germinal) mutation rate, that is, the frequency of errors in replicating DNA when producing a gamete. In the field of molecular anthropology, for example, microsatellite mutation rates are used to estimate the coalescence time between the alleles of a locus [4], the date of introduction/expansion of a variant in a population [5,6,7,8] and a variety of evolutionary events [9,10].

The main mechanism behind length mutations in microsatellites is thought to be the polymerase template slippage [11,12]. DNA strand slippage may transiently occur during DNA synthesis, which may result in mutant products where repeat units are added or deleted within the microsatellite [11,12,13]. Several factors influence microsatellite mutation rates, such as the (i) allele length, (ii) repetitive motif size and sequence, and (iii) parental age and gender. Indeed, (i) longer alleles tend to have higher mutation rates [14]; (ii) longer repeats tend to have lower mutation rates and, among those with the same length, mutations vary according to their sequences [15,16]; (iii) older individuals and males present higher mutation rates [17].

Mutations that involve the gain or loss of a single repeat in the transmitted parental allele single-step mutations are assumed to be preponderant over mutations involving the gain or loss of more than one repeat (multistep mutations) [18,19]. Indeed, the most accepted mutational model is the so-called stepwise mutation model (SMM), which considers single-step mutations as the most frequent when compared to multistep mutations [20]. This bias between single and multistep mutations is supported by studies on Y microsatellites, where length mutations are undoubtedly and specifically identified in simple structure markers [21,22,23]. Under the SMM framework, a single-step mutation is assumed to have occurred whenever this explains the genotypic incompatibility observed in a duo or trio familial configuration. The standard method for estimating microsatellite mutation rates is detecting and quantifying Mendelian incompatibility rates in one-generation family genotypic configurations, considering either the father or the mother and the child (the so-called duos), or considering both parents and the child (trios) [22,24,25,26,27]. However, except in the case of simple structure markers in haploid systems [21,28,29], this methodology entails the underestimation of mutation rates, as mutations may not necessarily lead to incompatibilities between parent(s) and child genotypes, originating ‘hidden’ or ‘covert’ mutations [29,30,31,32] (see Figure 1 for examples).

It is known that the likelihood of a mutation resulting in a Mendelian incompatibility is correlated with the type of familial configuration used [28,29], with biases being greater when duos, instead of trios, are analyzed [28,29,30,31,33]. A correction method was described previously [29], but the absence of an informatics tool for its implementation may have prevented its use.

In this work, we showed how the magnitude of the underestimation of autosomal microsatellite mutation rates varies with the populations’ allele frequency distribution spectrum, using simulated parent(s)–child duos and trios in different populational scenarios and assuming single-step mutations. Simulated familial duos and trios were generated considering a single-step mutation for each familial clustering and marker, using real and theoretical mock allele frequency distributions (henceforth called mock). Mock allele frequency distributions were considered to diversify the analyzed population allelic backgrounds and were designed by us considering pre-defined distributions. The populations/markers showing the highest rates of hidden mutations were assumed to be those with the greatest mutation rate biases when a standard approach to quantifying the Mendelian incompatibilities (Mendelian incompatibilities approach (MIA)) is used.

We aim to study the magnitude and type of biases obtained in mutation rate estimates, depending on the population allele frequency distribution and the type of familial data that are considered.

The implications for molecular anthropology are highlighted and a simple framework to correct the naïf mutation estimates, obtained through an analysis of Mendelian incompatibilities, is presented, along with a user-friendly and freely available informatics tool for the corresponding correction of incompatibility rates to mutation rates.

## 2. Materials and Methods

Familial genotypic configurations, mother–child or father–child duos, and mother–father–child trios, were generated by resorting to Python™ programming language. Since we aimed to measure the proportion of hidden mutations present in different population backgrounds, parental genotypes were randomly attributed from both real and mock population allele frequency distributions. Real allelic distributions concern ten autosomal microsatellites: CSF1PO, D1S1656, D21S11, D2S441, D3S1358, FGA, SE33, TH01, TPOX, and VWA, for the Norway, Somalia, and Spain populations [34]—see Appendix A for a graphic representation of the allelic distributions and Appendix A for population information (size, expected heterozygosity, polymorphism informative content (PIC) allele number, and allelic range). These markers were selected due to their distinct allelic distributions. On the other hand, to diversify the scenarios obtained from real populations and forensic markers, six mock (predefined) frequency distributions were designed by us: normal, bimodal, and constant distributions. For each case, narrow and wide allelic spans (with 10 and 20 alleles, respectively) were considered (Figure 2).

Parental genotypic configurations were generated by considering the allele frequencies of Norway, Somalia, and Spain populations [34] and the allele frequencies of the mock distributions (Figure 2). For each marker and population database, 1,000,000 familial genotypic configurations (duos and trios) were simulated, assuming, for each case, the occurrence of exactly one single-step mutation. To simulate the parental alleles, a cumulative relative frequency associated with each allele was considered for each marker. The two and four parental alleles considered in the case of duos and trios, respectively, were obtained considering two and four, resp., random numbers between 0 and 1. The corresponding allele was then chosen considering this number; the greater the frequency of the allele, the most likely that the allele was selected. When trios were simulated, the meiosis suffering mutation (either paternal or maternal) was randomly selected. Mutated alleles were assumed to be transmitted to the offspring, while the other filial allele was randomly selected either from the population (in the case of duos) or the other parent (in the case of trios).

As a parental mutated allele was assumed to be transmitted to the offspring in all the cases, there were two possible outcomes for each simulated familial configuration: the familial genotypic configuration was incompatible with the Mendelian inheritance, or otherwise. Under the standard, general, approach, the rate of the cases resulting in the first for a specific marker would be presented as the marker specific average mutation rate; while the rate of cases that resulted in the latter equates to the rate of hidden mutations, which would remain unnoticed.

The rate of hidden mutations was quantified for the different markers, populations, and familial configurations, assuming that the higher the rate, the greater the bias of the corresponding marker-specific mutation rate estimated through Mendelian incompatibilities.

Linear regression analyses were performed using Microsoft Excel^®^. The heterozygosity of each marker was calculated as $1-\sum_{i}p_{i}^{2},{\;\mathrm{where}\;p}_{i}\;\mathrm{is}\;\mathrm{the}\;\mathrm{frequency}\;\mathrm{of}\;\mathrm{the}\;\mathrm{allele}\;i.$

Fisher Exact tests to ascertain p values were computed considering a level of significance equal to 0.05. Python algorithms are openly available at https://github.com/econdesousa/Incomp2Mut.git (accessed on 23 March 2022), along with an informatics tool.

To replicate the simulations described in this work, and obtain a corrected mutation rate estimate for any marker, population, or familial configuration, the algorithms in https://github.com/econdesousa/Incomp2Mut.git (accessed on 23 March 2022) must be run for the target marker and incompatibility rates obtained through MIA. An example file on how to present the allelic distributions and a detailed explanation in video format of how to proceed with the informatic analyses are provided.

## 3. Results

In this section, results are presented comparing the accuracy of the estimates obtained for autosomal STR mutation rates, considering the analysis of both familial duos and trios, for different markers, populations, and mock allele frequencies, through the evaluation of the hidden mutation rates that were observed—see Table 1 and Figure 3.

### 3.1. Real Allele Frequency Distributions

As expected, parent–child duos concealed single-step mutations more often than parent–child trios (Table 1). These biases were strongly dependent on the allele frequency distribution of the marker. As a striking example, when considering the population of Norway, the rate of hidden mutations obtained for trios in marker CSF1PO were greater than the one obtained for marker D1S1656 when considering duos. This indicates that, for that population and markers, better estimates are expected for the D1S1656 mutation rate when duos are studied than for CSF1PO when trios are used once the same number of meiosis are analyzed. Indeed, within each of the three considered population databases [34], widely different proportions of hidden mutations were found for the 10 analyzed markers, even when the same familial configurations were used (either duos or trios). For example, in parent–child duos, considering the Norwegian database, mutations were concealed 4.3 more often for TPOX than for SE33. Globally, the proportion of hidden mutations varied from 5.4% (in SE33, for the Norwegian population, using trios) to 62.2% (in TPOX, also for the Norwegian population, when using duos). The ratio of hidden mutations found for the 10 analyzed markers in each population was computed. Statistically significant differences were found for virtually all pairwise comparisons; see Appendix A. Within all populations, the standard deviation in the ratios of marker-specific hidden mutations was shown to be high, varying between 0.058 (for Somalia, in trios), and 0.142 (for Norway, in duos); see Appendix A. Our results show that distinct levels of confidence for mutation rate estimates are expected for different markers within the same population; thus, marker-specific mutation rates should be estimated.

Furthermore, marker-specific pairwise analyses were computed, comparing the proportion of hidden mutations that were obtained for the different populations we studied; see Appendix A. Most (98.6%, α = 0.000115) pairwise population comparisons showed that the ratio of hidden mutations obtained for a specific marker significantly differed from population to population.

Therefore, we conclude that the difference between incompatibility rates (obtained through observations of Mendelian incompatibilities in duos or trios) and mutation rates depends on the allele frequency distribution in the population, whatever the familial configuration used (duos or trios). Globally, the markers that showed the worst and best mutation rate estimates (highest and smallest rates of hidden mutations, respectively) were D3S1358 and SE33, respectively.

None of the three populations we analyzed showed a consistent lowest value of hidden mutations across all markers, with most markers showing statistically significant pairwise differences for the rate of hidden mutations when analyzed in different populations; see Supplementary Material File S2. Nevertheless, the standard deviation associated with the marker-specific hidden mutation rates analyzed in different populations was small (maximum average σ = 0.031, see Appendix A), in contrast with the one found when different markers were analyzed within a specific population (maximum average σ = 0.124; see Appendix A).

### 3.2. Mock Allele Frequency Distributions

The six mock-allelic distributions: normal (narrow and wide), bimodal (narrow and wide), and constant (narrow and wide), showed widely different proportions of hidden mutations. More specifically, in parent–child duos simulated considering the normal distribution and 10 alleles, mutations were concealed 4.4 more times than in trios. Globally, the proportion of hidden mutations varied from 52.0% (in the wide constant distribution, using trios) to 69.6% (in the narrow normal distribution, using duos). As before, our results show that distinct levels of confidence for mutation rates estimates are expected for markers with different allelic distributions, again with duos hiding more mutations than trios. Besides, markers with a narrower distribution, i.e., with fewer alleles, hid more mutations than markers with more alleles. The ratio of hidden mutations was greater when the number of alleles increased for all the analyzed distributions. For example, for a normal, unimodal distribution with 10 markers (narrow distribution), duos concealed 4.4 times more mutations than trios. This figure increased to 6.8 times when 20 alleles were considered. This shows that biases resulting from the analysis of duos for estimating mutation rates via the computation of Mendelian incompatibility rates may be greater in less polymorphic populations. There is also a linear correlation between the expected heterozygosity and the rate of hidden mutations observed for the six mock allelic distributions for both duos and trios (r^2^ = 0.9841 and r^2^ = 0.9912, respectively); see Figure 4. Nevertheless, for the real allelic distributions that were studied, this high correlation was only verified for the case of duos (r^2^ = 0.911; and r^2^ = 0.4609 for trios).

## 4. Discussion

The accuracy of autosomal mutation rate estimates obtained through Mendelian incompatibilities varies between markers and populations according to allele frequency distributions, whatever the type of one-generation family data (parent(s)–child duos or trios) employed. Since mutations do not necessarily lead to Mendelian incompatibilities, this approach inherently underestimates their frequency.

It was previously acknowledged that the mutation rate estimates obtained through the observation of Mendelian incompatibilities at autosomal and X-chromosomal transmissions imply biases [28,29,30,31,33], and some procedures to correct them were already published for autosomal transmission [29,30,31,33]. Despite this, the generally accepted approach continues to be the direct estimation of mutation rates through the counting of Mendelian incompatibilities, without any correction; see, for example [24,25,26,27,35,36,37,38].

In autosomes, biases are more important in parent–child duos than in trios [29], showing that pooling data from the two types of sources is not acceptable, as it prevents any kind of a posteriori correction.

We have demonstrated that the probability of the occurrence of hidden mutations depends on the allele frequency distribution; therefore, the same marker may show different estimates in distinct populations despite the mutation rate value being the same. At this point, it should be highlighted that the real frequency distributions available for our analyses correspond to the markers designed for forensic individual identification, comprising STRs with a high diversity within, rather than between, populations. This is not the case for markers of anthropological interest, which were selected to maximize the differences between populations. In this case, pooling data from different populations to estimate mutation rates should be carefully thought out and planned, as different allelic distributions carry different likelihoods of disclosing mutations.

The framework we present in this work, described in the Materials and Methods section and thoroughly explained in https://github.com/econdesousa/Incomp2Mut.git (accessed on 23 March 2022), can be used to correct the mutation rates estimated through the MIA. To obtain the proposed corrective factor, it is only necessary to know the distribution of the allele frequencies at the loci of interest, the incompatibility rate observed and whether duos or trios were used to ascertain said rate. The output will be the corrected (for hidden mutations) mutation rate for the analyzed microsatellite. As exemplified, if parent–child duos are used and an R rate of Mendelian incompatibilities is found at the CSF1PO locus in the Norwegian population, the corrected value of R/(1–0.546) should be used as the estimated mutation rate at this locus and population, which represents nearly double the value estimated via MIA.

## 5. Conclusions

The accuracy of microsatellite mutation rate estimates obtained through the observation of Mendelian incompatibilities in parent(s)–child duos or trios depends on several factors, including the population allele frequency distribution. We showed that even when trios are used, as many as 27.1% (as obtained for marker D3S1358 for the population of Spain) of the mutations did not lead to any incompatibility. Although we framed our analyses under the knowledge that single-step mutations are the most frequent, the magnitude and the types of the biases increase if other mutations are considered (data not shown). We also did not consider the causes of the evidenced differences in the estimates across populations, i.e., whether biases are due to simple statistical properties of allelic distributions or intrinsic differences in allelic mutability. Whatever the reasons for these differences, its effect is the same on the estimation accuracy, as it only depends on the allele frequency distribution. However, the long-term evolutionary consequences are different. It is also important to note that population differentiation is expected to be lower for autosomal than for heterosomal markers, which are more susceptible to genetic drift. Therefore, the variation observed in the frequency of hidden mutations among populations when using autosomal markers may be even higher for heterosomal microsatellites.

The impact of this systematic underestimation inherent to the approach can be particularly burdensome in crucial anthropological problems, such as when dating evolutionary events; see, for example, [6,7,10].

We propose a simple method to obtain mutation rate estimates from Mendelian incompatibilities when using familial duos or trios, aiming to minimize the impact of hidden mutations. An informatics tool is provided at https://github.com/econdesousa/Incomp2Mut.git (accessed on 23 March 2022) to replicate this approach and obtain corrected mutation rates for any autosomal microsatellite, employing the allele frequency distribution and incompatibility rate estimated for either duos or trios.

## Figures and Tables

**Figure 1 genes-13-01248-f001:**
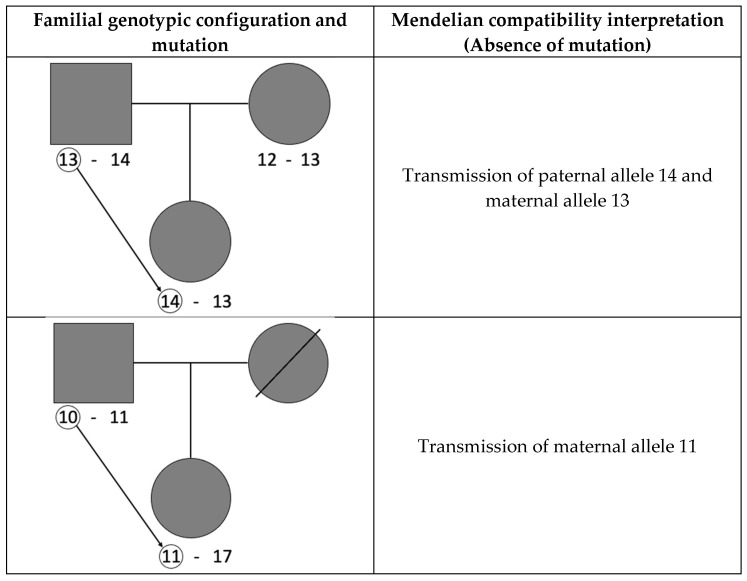
Examples of hidden mutations occurred in a parent–child duo and a parents–child trio, at an autosomal microsatellite. The arrows and circles indicate the allele transmission involving a mutation.

**Figure 2 genes-13-01248-f002:**
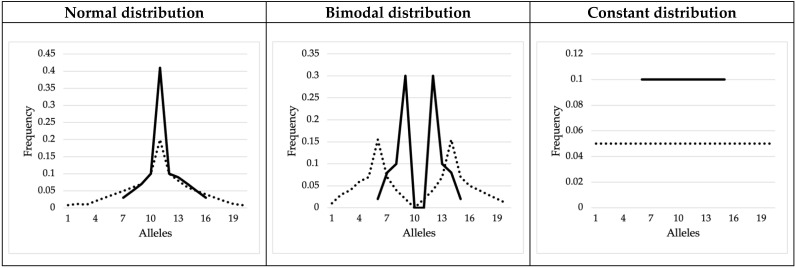
Mock allele frequencies, considering predefined distributions. The full lines correspond to the situation where 10 alleles are considered and the dotted lines to the cases with 20 alleles (narrow and wide distributions, respectively).

**Figure 3 genes-13-01248-f003:**
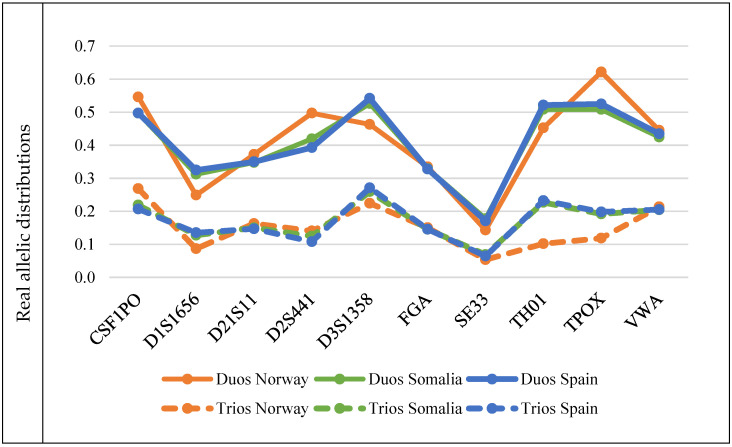
Graphical representations of the proportion of hidden mutations per marker (upper for real population distributions and lower for mock ones, considering the indicated distribution) and familial configuration. Full lines connect the dots corresponding to the proportion of hidden mutations for each marker in duos; dotted lines connect the dots corresponding to the proportion of hidden mutations for each marker in trios. For the mock distributions, N refers to the number of alleles considered in the marker. For example, “Normal (N = 10)” refers to the mock marker designed with a normal and narrow distribution, with 10 alleles, whereas “Normal (N = 20)” refers to the mock marker designed with a normal and wider distribution, with 20 alleles.

**Figure 4 genes-13-01248-f004:**
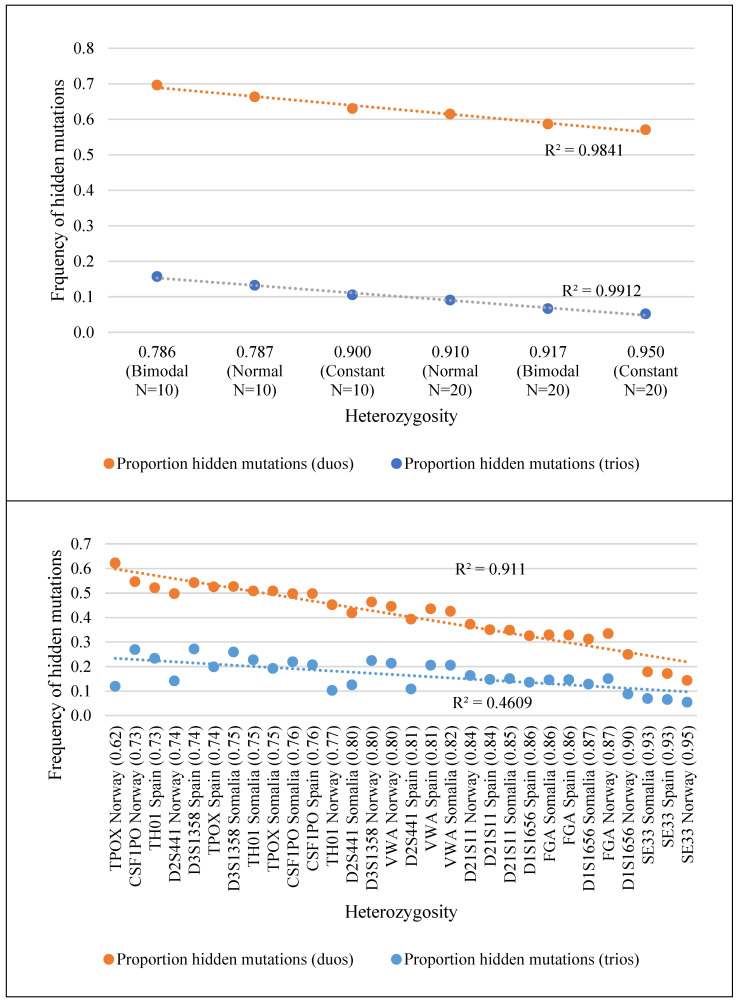
Graphical representations of the correlation between the frequency of hidden mutations observed per marker (upper for real population distributions and lower for mock ones, considering the indicated distribution) and markers’ heterozygosity. Orange corresponds to the correlation of hidden mutations and the heterozygosity of each marker in duos, and blue in trios. For the mock distributions, N refers to the number of alleles considered in the marker. For example, “Normal (N = 10)” refers to the mock marker designed with a normal and narrow distribution, with 10 alleles, whereas “Normal (N = 20)” refers to the mock marker designed with a normal and wider distribution, with 20 alleles. Heterozygosity was calculated as $1-\sum_{i}p_{i}^{2},{\;\mathrm{where}\;p}_{i}\;\mathrm{is}\;\mathrm{the}\;\mathrm{frequency}\;\mathrm{of}\;\mathrm{the}\;\mathrm{allele}\;i$.

**Table 1 genes-13-01248-t001:** Rates of hidden mutations per marker (real and mock allelic distributions) and familial configuration considered (either duos or trios). One single-step mutation was simulated in one, randomly selected, parental meiosis of each of the 1,000,000 parent–child duos and parents–child trios, considering the allelic distributions of 10 autosomal STRs for the populations of Norway, Somalia, and Spain [34] and the allelic distributions of the 6 artificially generated markers. In the latter, N refers to the number of alleles in the marker.

Markers			Duos			Trios		
			Norway	Somalia	Spain	Norway	Somalia	Spain
**Allele Frequencies**	**Real allelic distributions**	**CSF1PO**	**0.546**	**0.497**	**0.497**	**0.269**	**0.219**	**0.207**
		**D1S1656**	0.249	0.312	0.325	0.087	0.128	0.135
		**D21S11**	0.372	0.348	0.35	0.163	0.15	0.147
		**D2S441**	0.497	0.419	0.393	0.141	0.125	0.108
		**D3S1358**	0.463	0.526	0.542	0.224	0.259	0.271
		**FGA**	0.334	0.329	0.328	0.15	0.145	0.146
		**SE33**	0.143	0.178	0.171	0.054	0.069	0.065
		**TH01**	0.452	0.508	0.521	0.102	0.227	0.233
		**TPOX**	0.622	0.508	0.525	0.119	0.192	0.198
		**VWA**	0.445	0.425	0.435	0.213	0.205	0.205
	**Mock allelic distributions**	**Normal (N = 10)**	0.696			0.157		
		**Normal (N = 20)**	0.614			0.09		
		**Bimodal (N = 10)**	0.663			0.132		
		**Bimodal (N = 20)**	0.586			0.066		
		**Constant (N = 10)**	0.63			0.105		
		**Constant (N = 20)**	0.57			0.052		

## Data Availability

Not applicable.

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
