# Peer review of "Estimations of Mutation Rates Depend on Population Allele Frequency Distribution: The Case of Autosomal Microsatellites"

_genes, 2022, doi:10.3390/genes13071248_

Round 1

Reviewer 1 Report

The authors present a method for recalibration of mutation rates for microsatellite loci when considering the source of the data (duos and trios) and the distribution of allele frequencies.  The title should be modified as the wording is unclear; suggest changing the title to “Estimations of mutation rates depend on population …” and not hyphenating the word “frequency”.  The manuscript is littered with hyphenated words that detract from the work.

Introduction:

The authors should be consistent with how STR is annotated; i.e., “short tandem repeat” in the Abstract and “Short Tandem Repeat” in the Introduction.

It is unclear how citations 1-3 reflect work that has addressed a wide range of “problems” (perhaps questions?) in scientific fields such as those listed.  Subsequent sentences of the Introduction are disconnected, creating further lack of clarity. This section of the Introduction needs considerable revision.

How does the length of the longest uninterrupted repeat sequence contribute to the slippage process leading to mutations?

Line 50: It is unclear what “over the other” is referring to.

Line 52: It is unclear what “as the most frequent” is referring to.

It is unclear why the various models were introduced, as they are not referred to again in the manuscript.

Line 96: Should reintroduce the MIA acronym.  Subsequent usage (for example, line 264) reverted back to Mendelian incompatibilities approach.  Be consistent.

Overall, the flow and connectivity of the Introduction is choppy and should be revised.

Results:

The two tables in Appendix 2 need legends that provided a detailed explanation of the presented data.

Line 191: It is unclear how the authors calculated the 99.9% value; assuming they are referring to the data in Appendix 2.

Line 194: What does “estimates’ accuracy” mean?

Discussion & Conclusion:

The implications of the authors’ findings for molecular anthropology are discussed at a bare minimum; lines 257-261 and 291-293.  As a result, this paper is less about anthropology and more about the assessment of mutation rates when considering familial configurations and allele distributions.  Therefore, either the discussion should be expanded or the focus narrowed.

The first sentence of the last paragraph of the Conclusions section is unclear.

Is the informatics tool "provided" or simply discussed?  This should be clarified.

Reviewer 2 Report

Dear authors, the manuscript is interesting, and the study falls into the scope of Genes. However, before this can be done, a minor revision should be made as specified below.

- Please, check the figure 1. Was it not supposed to the arrows and circles indicate the same allele in the transmission between parent-child?

- Lines 103-104: “TPOX, and 103 VWA,”

- Line 107: “allele number, and allelic range”
